# Gaia BH1: A Key for Understanding the Demography of Low-q Binaries in the Milky Way Galaxy

Oleg Malkov 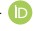

Institute of Astronomy of RAS, Moscow 119017, Russia; malkov@inasan.ru; Tel.: +7-495-951-7993

**Abstract:** The recently discovered Gaia BH1 binary system, a Sun-like star and a dark object (presumably a black hole), may significantly change our understanding of the population of binaries. The paper presents the components mass ratio ($q$) distributions of binary systems of different observational classes. They all show a significant shortage of low-q systems. In this work, I demonstrate (quantitatively) how our ignorance extends, and point out the importance of discovering and studying systems like Gaia BH1. In addition, an approximate mass–temperature relation and mass ratio–magnitude difference relation for main-sequence stars are presented here.

**Keywords:** binary stars; orbital; eclipsing; spectroscopic; mass ratio distribution; Gaia

## 1. Introduction

In [1], the discovery of binary system Gaia DR3 4373465352415301632 is reported with unusual components. It is a bright, nearby Sun-like star ($T_{\text{eff}}$ = 5850 K, bf log g (cm/s$^2$) = 4.5) with mass $m_2 = 0.93 \pm 0.05$ (hereafter, masses are given in solar mass, $M_\odot$), orbiting a dark object with mass $m_1 = 9.62 \pm 0.18$. The authors [1] believe that the dark object is a black hole, have given it the designation Gaia BH1 and admit that the origin of the system is uncertain. In [2], a boson star was proposed as an explanation of the nature of the dark object. It was assumed that this object really consists of a black hole and a Sun-like star, that there is nothing unusual about the origin of this binary system, and that Gaia BH1 just exhibits a rather small value of $q$ (the mass ratio of the components), which was even smaller in the epoch when the black hole was a massive star. Below, we show the importance of such objects in our understanding of low-q systems demography.

## 2. Mass Ratio Distribution of Binaries

The statistics of binary systems with small component mass ratio q ($=M_2/M_1$), so-called low-$q$ binaries, are poorly understood. Obviously, the components of such systems have, as a rule, very different luminosities (and the black hole or neutron star does not contribute to the light at all). This high brightness contrast prevents the detections of such systems by astrometric, interferometric, photometric, spectroscopic, and other methods. This strong selection effect leads to the fact that modern catalogs and databases of binary stars of different observational types contain predominantly high-$q$ binaries, while high-contrast systems remain undetected.

Below, we analyze the content of the principal catalogs of binary stars of various observational classes. The main focus was on the observing techniques, which yield a large number of binary systems: visual, eclipsing, and spectroscopic binaries. Indeed, our analysis of the main observational classes of binaries shows a relative abundance of high-$q$ and an obvious lack of low-$q$ systems.

## 2.1. Spectroscopic Binaries

To construct the q-distribution of spectroscopic binaries, we have taken 1642 SB2 systems from the most complete and representative source, the SB9 catalog [3]. Their $K_1/K_2$ distribution (which is, in fact, the $q$-distribution) is represented by the blue histogram in Figure 1. Here, $K_1$ and $K_2$ are velocity amplitudes of the primary and secondary components, respectively.

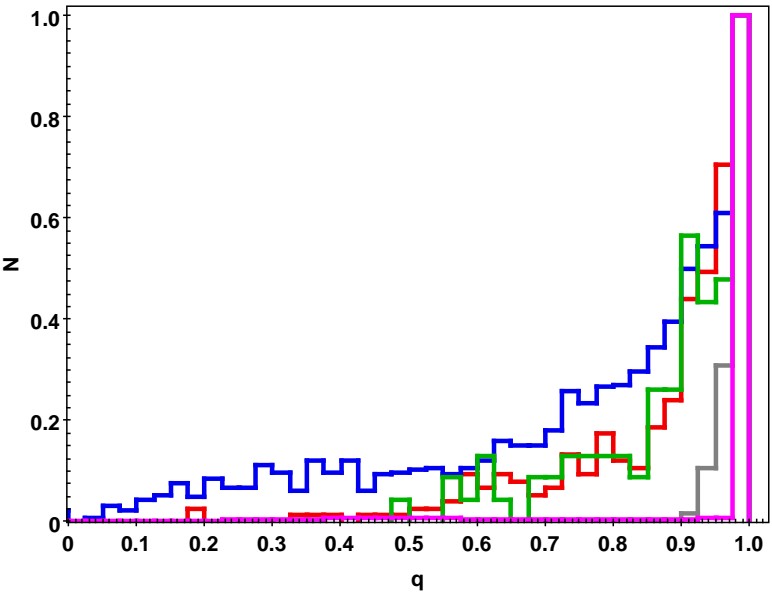

**Figure 1.** q-distribution of spectroscopic SB2 (blue, see Section 2.1), orbital (red, see Section 2.2), DLEB (green, see Section 2.3), detached main-sequence eclipsing (gray, see Section 2.4), and Gaia DR3 (magenta, see Section 2.5) binaries. The histograms are normalized to the maximum.

It should be noted that several previous studies modeled the detection biases of their approaches (in particular towards low-q early-type spectroscopic systems), all demonstrating that the detection bias against low-q systems is high [4–7].

## 2.2. Orbital Binaries

Kepler's Third Law allows one to calculate the total mass (the so-called dynamical mass) of a binary system, for which the period, semi-major axis and trigonometric parallax are known (this information is usually available for orbital binaries, i.e., visual binaries with orbital solution). Alternatively, individual masses of (at least main-sequence) components (so-called photometric mass) can be calculated from their visual brightness, trigonometric parallax, interstellar extinction, and the mass–luminosity relation. This was performed by [8] for 3350 objects from the most complete and representative source, the sixth catalog of orbits of visual binary stars, ORB6 [9]; of which 326 of them demonstrate a decent agreement between dynamical and photometric masses. Their $q$-distribution is represented by the red histogram in Figure 1.

## 2.3. Double-Lined Eclipsing Binaries

Components of detached double-lined eclipsing binaries (DLEBs) satisfy the criterion that their mass and radius are known to $\pm 3\%$ or better. Study [10] makes one of the best critical compilations of 95 DLEB systems. Their $q$-distribution is represented by the green histogram in Figure 1.

### 2.4. Detached Main-Sequence Eclipsing Binaries

It is impossible to estimate the mass ratio of the components of an eclipsing binary without additional spectroscopic or astrometric observations. However, it is possible to estimate the effective temperature ratio of the components. Study [11] proposed the following relation:

$$\left(\frac{T_{\text{eff},1}}{T_{\text{eff},2}}\right)^4 \approx \frac{1 + 0.4A_1}{1 + 0.4A_2} \tag{1}$$

where $T_{\text{eff},i}$ are effective temperature of the components, and $A_1$, $A_2$ are depths of the primary and secondary minima at the primary and secondary eclipse, respectively.

To transfer the temperature ratio to the mass ratio, one can use well-studied DLEB objects, mentioned above, with highly accurate parameter values. Mass and effective temperature of DLEB components from [10] are shown in Figure 2. Three giants and components of the least massive system in the list, CM Dra, were removed from the set. Linear approximation (a least-squares best fit) was performed in logarithmic scale:

$$\log T_{\text{eff}} = 0.645 \log(M/M_\odot) + 3.730, \tag{2}$$

with the correlation coefficient being 0.989 (a more detailed mass-temperature relation can be found, e.g., in [12]). This allows us to derive $q$ from the $T_{\text{eff}}$ ratio:

$$\log q = \frac{1}{0.645} \log\left(\frac{T_{\text{eff},2}}{T_{\text{eff},1}}\right). \tag{3}$$

To construct a q-distribution for eclipsing binaries, we used objects from one of the most complete and representative source, the catalog of eclipsing variables, CEV [13]. Relation (2) is valid only for main-sequence stars, so we have selected only detached main-sequence (DM) systems from the CEV. We selected systems that have a DM evolutionary class published in the literature or assigned as a result of our own classification [14]. For 973 of them, the depths of both minima are known, and the procedure of $q$ estimation was performed for them.

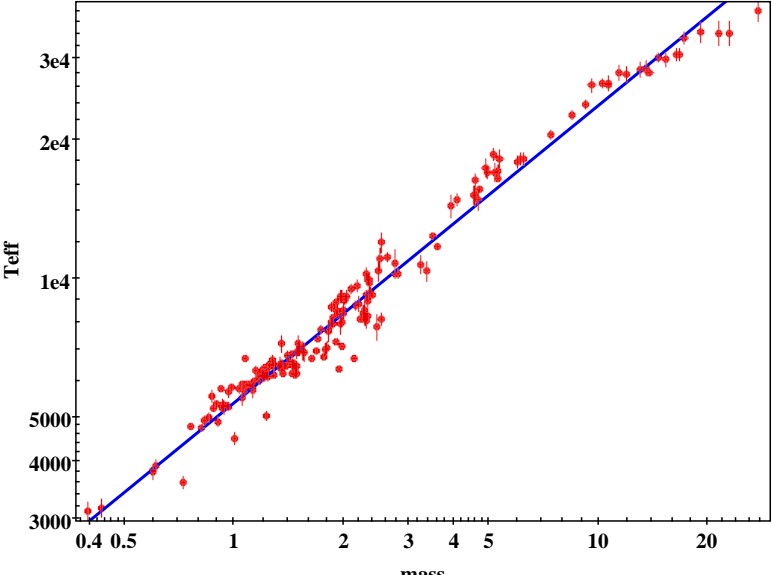

**Figure 2.** Mass ($M/M_\odot$) vs. $T_{\text{eff}}$, K of DLEB [10]. The blue line is a linear approximation.

### 2.5. Gaia DR3 Non-Single Stars

The Gaia DR3 archive [15] provides access to a table containing non-single star (NSS) orbital models for sources compatible with an orbital two-body solution (NSS two body orbit[1]). This covers astrometric binaries, spectroscopic binaries, eclipsing binaries and certain combinations thereof. The table was cross-matched with a table of masses derived from the NSS solutions with orbital parameters in the DR3 NSS two-body orbit table (Binary masses[2]). The resulting table contains 198,880 binaries.

Masses of both components are derived here for three observational types of binaries: Orbital+SB2 (visual binaries with known orbital elements also observed spectroscopically, with lines of both components in the spectrum), EclipsingSpectroSB2 (combined eclipsing binary + spectroscopic orbital model), and Eclipsing+SB2 (double-lined eclipsing binaries, see Section 2.3). In all these cases, the mass of the primary component was derived directly from the NSS solutions. These three observational types are represented by 46, 3, and 109 binary systems, respectively. Their $q$-distribution is shown by the magenta histogram in Figure 1. Other cases are:

| | |
|---|---|
| Orbital + M1 | 113,246 binaries |
| SB1 (Single Lined Spectroscopic binary model) + M1 | 60,474 binaries |
| AstroSpectroSB1 (Combined astrometric + single lined spectroscopic orbital model) + M1 | 17,646 binaries |
| SB2(Double Lined Spectroscopic binary model) + M1 | 3945 binaries |
| Orbital + SB1 + M1 | 3026 binaries |
| Eclipsing + SB1 + M1 | 311 binaries |
| EclipsingSpectro (Combined eclipsing binary + spectroscopic orbital model) + M1 | 74 binaries |

In all these cases, "+M1" means that the mass of the primary component is the input mass from isochrone fitting, see Appendix D of [16].

Note that further investigation of the Gaia data on binary stars seems quite promising. Thus, ref. [17], studying candidate Gaia DR3 compact object companions, found that the distribution of *inferred* mass ratios shows a large number of low-$q$ systems (see their Figure 17).

### 2.6. Interferometric Binaries

The masses of binary stars cannot be determined from interferometric observations, but it is often possible to estimate the magnitude difference, $dm$, or brightness ratio, $b_1/b_2$, of the components. In particular, the $dm$ or $b_1/b_2$ values are contained in the catalogs/lists presented in Table 1: Balega+ [18], CHARM2 [19], Strakhov+ [20].

**Table 1.** Catalogs/lists of interferometric binaries.

| Catalog | $N_{dm}$ | $N_{0.1}$ | Star | $b_1/b_2$ | $dm$ | SpT | $q$ |
|---|---|---|---|---|---|---|---|
| Balega+ | 111 | 0 | $\theta$ Ori A | | 3.23 | B0V | 0.45 |
| CHARM2 | 313 | 0 | $\rho$ Ari | 23.8 | 3.44 | A3V | 0.23 |
| Strakhov+ | 372 | 1 | HD 340178 | 315.5 | 6.25 | A3 | 0.09 |

$N_{dm}$ is the number of $dm$-measurements, $N_{0.1}$ is an estimated number of binaries with mass ratio $q \leq 0.1$, *Star* is the cataloged main sequence star (earlier than mid-A) with the largest brightness ratio; its $b_1/b_2$, $dm$, spectral type, and $q$ (estimated from Figure 3) are also provided.

Unfortunately, due to the non-linearity of the mass-luminosity relation (MLR), it is not possible to unambiguously match the values of $dm$ and $q$. But, such relations can be obtained by fixing the mass (or luminosity) of one of the components. Figure 3 shows the relations for the fixed luminosity of the main component. The primary magnitude $M_{V1}$ takes values 2, 1, 0, $-1$, $-2$, $-3$, $-4$, $-5$ mag, which correspond approximately to the main-sequence spectral types A5, A1, B8, B4, B2.5, B1.5, B0, O8, respectively (the spectral types were estimated according to [21]). The luminosity of the secondary varies from $M_{V1}$

to 9.0 mag (which corresponds approximately to M0V). The MLR from [12] was used in the calculations:

$$\log M(M_V) = 0.525 - 0.147 M_V + 0.00737 M_V^2, \tag{4}$$

which is valid for $-5.0 < M_V < 9.0$ mag.

By estimating the $dm$ values for systems whose component mass ratios can reach $q = 0.1$ and less from Figure 3, it can be seen that the difference in the magnitudes of the components $dm$ should be at least $9.0^m$ (for mid-A primaries) or $5.5^m$ (for late-O primaries), which, according to

$$dm \equiv m_2 - m_1 = 2.5 \log(b_1/b_2) \tag{5}$$

or

$$b_1/b_2 = 10^{0.4 dm}, \tag{6}$$

correspond to the brightness ratio $b_1/b_2 \approx 4000$ and 160, respectively. The analysis of the interferometric catalogs mentioned above shows that the fraction of objects with such significant $dm$ is vanishingly small (see the $N_{0.1}$ value in Table 1). Table 1 also contains information on the main sequence star in the given catalog, which has the largest observed brightness ratio and which is earlier than mid-A. Thus, the results of modern interferometric observations also contain very poor data on low-q binaries. Note, however, that in their recent study of O-type stars with optical long baseline interferometry [22] resolved 19 companions in 17 different systems, five of which have estimated $q \approx 0.1$, but there is a recent study of nearby hierarchical systems listing a number of low-$q$ binaries, including several below $q = 0.1$ [23].

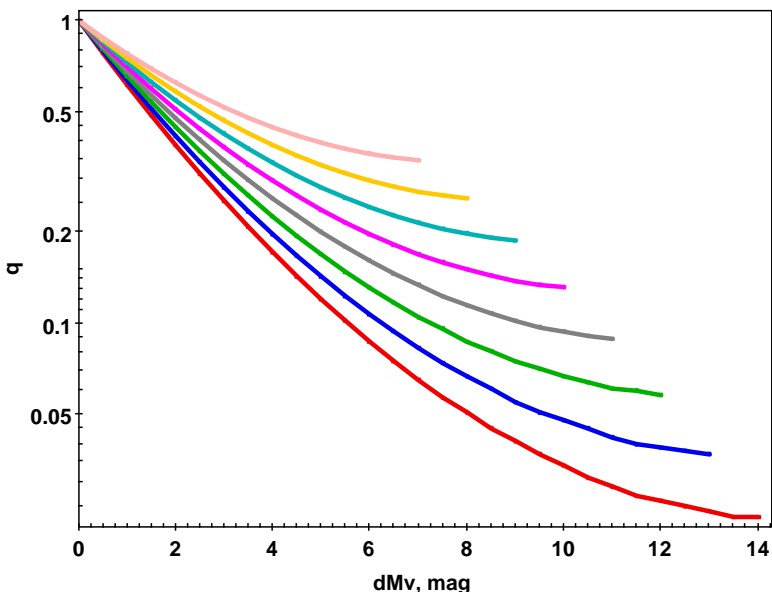

**Figure 3.** Mass ratio: magnitude difference relation for main-sequence binaries. Different curves represent relations for the following primary magnitudes $M_{V1}$ (from top to bottom): 2, 1, 0, $-1$, $-2$, $-3$, $-4$, $-5$ mag. It corresponds, approximately, to main-sequence spectral types A5, A1, B8, B4, B2.5, B1.5, B0, O8 [21], respectively. Secondary magnitude $M_{V2}$ varies from $M_{V1}$ to 9.0 mag (M0V). Note that there is a logarithmic scale for the Y-axis.

We have discussed several observational types of binary systems above, and consider the most "representative" observational types, which are those for which at least hundreds of binary systems are known (the exception is DLEB, see Section 2.3, which is included due to the high accuracy of the derived parameters). However, it is also necessary to mention the high-contrast imaging method. The number of binaries of this observational type is not yet very large; however, this method makes it possible to detect systems with a moderate and small value of the mass ratio of the components $q$ [24–26].

The resulting q-distributions are shown in Figure 1. The figure does not reflect the real *q*-distributions of binary systems, it only demonstrates the degrees of our ignorance of the real situation in the low-*q* systems for binaries of various observational classes. It can be seen from Figure 1 that, when studying systems with $0.1 \leq q \leq 0.5$, we can only count on the spectroscopic binaries, i.e., we are limited to relatively bright stars. The most dramatic situation is for Gaia NSSs and detached main-sequence eclipsing binaries, where we can analyze only systems with $q > 0.9$. It seems promising to combine data from the (very representative catalog) NSS (see Section 2.5) with a source of accurate data on spectroscopic binaries SB9 (see Section 2.1). Preliminary results of such a study can be found, for example, in [27].

It should be added that the distribution of binary systems by the mass ratio of the components (which is relatively easily obtained from observations) for main sequence [28] and pre-main sequence [29] stars is a perfect tool for determining the shape of the stellar initial mass function (IMF), which cannot be observed directly and should be estimated from the indirect techniques [30] (see also the recent study on the relationship of IMF, pairing function and q-distribution [31]).

### 3. Gaia BH1—A Rare Low-q Binary

In Section 2, it was shown that we have a notable lack of information about low-q systems. From this point of view, the Gaia BH1 is a very valuable finding, the importance of which cannot be underestimated. Indeed, the mass of the black hole indicates that the initial mass of the massive component, according to the remnant mass versus initial stellar mass relation, was (depending on the details of the presupernova evolution of massive stars, especially relating to convection and mass loss) $M_{1,\text{ini}}$ = 23 to 27 $M_\odot$ [32,33]. This means that mass ratio of the pre-Gaia BH1 object was $q$ = 0.03–0.04, a value completely unattainable with current observations of binary stars.

Note that there exist dozens of black-hole low-mass X-ray binaries consisting of a low-mass donor star (with a mass less than 1 $M_\odot$) and a black-hole accretor (with a mass of $\approx 10 M_\odot$). Certainly, mass transfer exists between two components. However, in our study, we consider binary systems demonstrating *initial q*-distribution, i.e., without mass transfer. The orbital period $P_{\text{orb}}$ = 185.6 days, and modest eccentricity $e$ = 0.45 of Gaia BH1 exclude the binary mass transfer, now or in the past. Note that such an eccentricity is very similar to that of the eccentric ($e$ = 0.44) binary millisecond pulsar PSR J1903+0327 discovered in the Galactic plane [34]. It should be added that the recently discovered Gaia BH2 system ($\approx 1 M_\odot$ red giant and $8.9 \pm 0.3 M_\odot$ black hole) [35] is also quite interesting in the context of this study, although it has a larger $q$ = 0.1 value. Note also that Gaia-Hipparcos proper motion changes for nearby targets can detect small accelerations caused by objects with mass as low as Jupiter-like planets [36].

Therefore, it seems useful to continue the search for similar objects in the Gaia data archive. In particular, three low-q candidates can be recommended from the NSS two body orbit list mentioned in Section 2.5. These candidates, together with Gaia BH1, are presented in Table 2. In addition to the identifiers and component masses, Table 2 contains information on the NSS solution type (here "Orbital" means "Orbital model for an astrometric binary" and "AstroSpectroSB1" means "Combined astrometric + single lined spectroscopic orbital model"). This search will make it possible to accumulate the necessary statistics for analyzing low-*q* systems. In addition, it will be possible to learn to take into account the selection effects that distort the statistics of binary systems of different observational classes.

**Table 2.** Gaia BH1 and other candidates for low-q binaries.

| Source ID | NSS Solution Type | $M_1/M_\odot$ | $M_2/M_\odot$ |
|---|---|---|---|
| 4373465352415301632 (BH1) | Orbital | 0.95 | 12.81 |
| 1864406790238257536 | AstroSpectroSB1 | 2.40 | 20.08 |
| 3640889032890567040 | Orbital | 1.01 | 123.47 |
| 6281177228434199296 | Orbital | 0.95 | 11.91 |

**Funding:** This research was funded by the Ministry of Science and Higher Education of the Russian Federation, according to the research project 13.2251.21.0177 (075-15-2022-1228).

**Data Availability Statement:** The data presented in this study are available on request from the corresponding author.

**Acknowledgments:** I warmly thank Annemarie Bridges and James Wicker for their great help in preparing the manuscript. The author thanks Dana Kovaleva and Boris Safonov for the valuable remarks and suggestions. I also thank my reviewers, whose constructive comments helped me to improve the paper significantly.

**Conflicts of Interest:** The author declares no conflict of interest.

## Abbreviations

The following abbreviations are used in this manuscript:

| | |
|---|---|
| SB9 | Ninth Catalog of Spectroscopic Binary Orbits |
| ORB6 | Sixth Catalog of Orbits of Visual Binary Stars |
| DLEB | Double lined eclipsing binaries |
| CEV | Catalog of eclipsing variables |
| DM | Detached main-sequence |
| NSS | Non-single star |
| INT4 | Fourth Catalog of Interferometric Measurements of Binary Stars |
| CHARM2 | An updated Catalog of High Angular Resolution Measurements |
| MLR | Mass–luminosity relation |
| IMF | Initial mass function |

## Notes

1    https://gea.esac.esa.int/archive/documentation/GDR3/Gaia_archive/chap_datamodel/sec_dm_non--single_stars_tables/ssec_dm_nss_two_body_orbit.html, accessed on 8 August 2023.

2    https://gea.esac.esa.int/archive/documentation/GDR3/Gaia_archive/chap_datamodel/sec_dm_performance_verification/ssec_dm_binary_masses.html, accessed on 8 August 2023.

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
