# Peer review of "Gaia BH1: A Key for Understanding the Demography of Low-q Binaries in the Milky Way Galaxy"

_galaxies, doi:10.3390/galaxies11050098_

Round 1

Reviewer 1 Report

Reviewer Report 

I have carefully read the manuscript entitled "Gaia BH1: a key for understanding the demography of low-q binaries in the Milky Way Galaxy" by Oleg Malkov.

Recently, it discovered Gaia BH1 binary system consisting of a Sun-like star and a dark object (presumably a black hole). This source may significantly change our understanding of the population of low-q binaries. The manuscript summarizes the components mass ratio distributions of binary systems of different observational classes, which indicate a significant shortage of low-q systems. This shortage can be filled with the help of search and study of objects like Gaia BH1. In my understanding, it is difficult to form low-q binaries in standard stellar and binary evolution theory, Therefore, this manuscript will raise some interests some colleagues in the astronomical community, and is appropriate and worthy of consideration for publication in the Galaxies. 

I would like to recommend the manuscript for the publication after the authors address the comments as follows.

Major comments:

1. How to define low-q binaries? In the Galaxy, there exist about a dozen black-hole (BH) low-mass X-ray binaries (LMXBs)   consisting of a low-mass donor star (with a mass less than 1 Msun) and a BH (with a mass of ~ 10 Msun). Certainly, a mass  transfer exists between two components. What is the difference between your defined low-q binaries and BH LMXBs?

2.You should present a detaled discussion on why it is difficult to form such a low-q system like Gaia BH1? For example,  a high eccentricity e = 0.45 can not be accounted for. Such an eccentricity is very similar to that of the eccentric (e = 0.44)   binary millisecond pulsar PSR J1903+0327 discovered in the Galactic plane (Champion et al. 2008, Science, 320, 1309), which  had challenged the standard binary evolution theory.  

3. Similar to point 2, you mention "This means that mass ratio of the pre – Gaia BH1 object was q = 0.03-0.04, a value completely unattainable with current observations of binary stars." in lines 139-141. Such a concluson is based on an assumption that the primordial binary does not experience a mass transfer? You should given a discussion on this point.

Minor comments:

1. In the title, Milky Way and Galaxy are a same meaning.

2. In equations (4)-(6), you use m_1 and m_2 to represent the magnitude of two components, however,  m_1 and m_2 were used to   represent the masses of two components.

3. Line 12, you mentioned "hereafter masses are given in solar mass". However, "m/m_sun" in figure 1,   "0.645 log(m/m_sun)" in equation (2), and m_1,ini = 23 to 27 m_sun in line 139.

4. Line 148, "Non-single-star (NSS)"->"NSS".

No.

Author Response

Dear reviewer,

I thank you for extremely useful and constructive comments,
they greatly helped me to improve the paper.
I have made all required corrections and responded
all questions raised. All revisions
are highlighted (boldfaced) in the text. The details
of the revisions are given below, point-by-point.

Please, note that corrections required by other reviewers
are also made and highlighted in the text.

I am ready to answer your questions if you still have them.
Thank you very much again for your cooperation,
With best regards,
Oleg Malkov

Major comments:

1. How to define low-q binaries? In the Galaxy, there exist about a dozen black-hole (BH) low-mass X-ray binaries (LMXBs)   consisting of a low-mass donor star (with a mass less than 1 Msun) and a BH (with a mass of ~ 10 Msun). Certainly, a mass  transfer exists between two components. What is the difference between your defined low-q binaries and BH LMXBs?

==> Thank you very much for pointing out this my omission. You are right, there are dozens LMXBs with low q.
However, in our study we consider binary systems demonstrating _initial_ q-distribution,
i.e., without mass transfer. I have added a couple of phrases in the last but one paragraph of Sect.3.

2. You should present a detaled discussion on why it is difficult to form such a low-q system like Gaia BH1? For example,  a high eccentricity e = 0.45 can not be accounted for. Such an eccentricity is very similar to that of the eccentric (e = 0.44)   binary millisecond pulsar PSR J1903+0327 discovered in the Galactic plane (Champion et al. 2008, Science, 320, 1309), which  had challenged the standard binary evolution theory.  

==> I am not saying that such low-q systems are difficult to form, I am only saying that
they are difficult to detect. Thank you for the useful reference on PSR J1903+0327,
I inserted it in the same paragraph.

3. Similar to point 2, you mention "This means that mass ratio of the pre - Gaia BH1 object was q = 0.03-0.04, a value completely unattainable with current observations of binary stars." in lines 139-141. Such a concluson is based on an assumption that the primordial binary does not experience a mass transfer? You should given a discussion on this point.

==> Yes, such a concluson concerns binaries which do not experience a mass transfer (see my answer to your point 1 above).

Minor comments:

1. In the title, Milky Way and Galaxy are a same meaning.

==> You are absolutely right. However, the guest editor of the Galaxies Special Issue
(where I submit my manuscript) recommended me to formulate the title in this way.
I have left these words in the title unchanged for now, but if you insist,
I will discuss with the guest editor the possibility of changing the title.

2. In equations (4)-(6), you use m_1 and m_2 to represent the magnitude of two components, however,  m_1 and m_2 were used to   represent the masses of two components.

==> Corrected. Now everywhere in the text the masses are denoted as 'M' (and not 'm').

3. Line 12, you mentioned "hereafter masses are given in solar mass". However, "m/m_sun" in figure 1,   "0.645 log(m/m_sun)" in equation (2), and m_1,ini = 23 to 27 m_sun in line 139.

==> It is easy for me to remove unnecessary "/m_sun" from the text.
However, in some cases it seems to me useful to reiterate that masses are given in solar mass.
For example, the reader may immediately be interested in figure 2 (in the previous version of the paper
it was figure 1), without paying attention to the warning about symbols at the beginning of the article.
Again, if you insist I will correct it.

4. Line 148, "Non-single-star (NSS)"->"NSS".

==> Corrected.

Reviewer 2 Report

Dear author,

please find attached my feedback on the manuscript titled "Gaia BH1: a key for understanding the demography of low-q binaries in the Milky Way Galaxy". I sorted my comments into general comments and specific ones to certain sections of the paper.

General:
-----------------------
- in my opinion, the title of the paper is misleading as most of the paper is not about Gaia BH1. The paper provides a compilation of literature studies, summarising the distribution of mass ratios determined by different authors and methods. The paper demonstrates that few low-q binaries are known, however, that does not help to further our understanding, neither of low-q binaries nor of Gaia BH1

- while it is interesting and insightful to provide a compilation of measured mass ratios, please note that this is very much what is expected: low-q systems of two main-sequence stars are by definition harder to detect as their masses, and therefore their light contribution is very different. This is therefore nothing new, as you state in the beginning of Sect. 2, and thus the result of the paper is not surprising.

- several previous studies modelled the detection biases of their approaches, in particular towards low-q systems. For example, this was done extensively for spectroscopic studies (for example Sana+2013, A&A 550, A107; Bodensteiner+2021, A&A 652, A70; Banyard+2022, A&A 658, A69; Shenar+2022, A&A 665, A148; all demonstrating that the detection bias against low-q systems is high)

- while listing several methods detecting binaries, it remains unclear which methods were included and which ones were not. For example, high-contrast imaging has proven to be able to detect low-q systems (for example Pauwels+2023, arxiv:2307.13500 and references therin). Please clarify why this technique was not included, or consider including it.

- along similar lines: for each of the methods, it remains unclear which catalogues / sources were considered and which ones were not. Please clarify this.

- concerning the figures: please sort them in a way that the first figure mentioned in the text is figure 1, and so on. Also, please include a legend (with the appropriate references when required) in each of the figures.

Specific:
-----------------------
Title: as mentioned above, I think the title should be changed to reflect the content of the paper better. Also, in my opinion, the 'q' should be written out in the title

Abstract: please introduce 'q' as mass ratio

Introduction:
- please define q already here, and not only in Sect. 2
- please provide the error bars for m2 quoted here
- please always put units on numbers (and please stick to the common convenvtions, for example using M_sun instead of m_sun

Section 2:
- line 23: for BHs, this sentence does not hold: while the mass ratio is high, the BH does not contribute to the light at all. Given that the introduction discussed BH1, please clarify this statement such that it makes this clear.
- in my opinion, it would help the reader in their understanding to expand the second paragraph of Sect. 2 to explain what will be done in the subsequent subsections of the paper. Here, it would also be good to provide more detail what 'the main observational classes' are, and how they are defined (see my comment above on high-contrast imaging)
- please provide more information about why which set of catalogue / data was used for each of the subsections. For example, why was the SB9 catalogue considered for the spectroscopic binaries?
- line 56: you mention 'depth of the primary and secondary minimum' - do you mean the primary and secondary eclipse? (the same holds for line 68)
- line 60: please clarify what exactly 'linear approximation' means, were the data including the error bars fitted? if yes, by which means (by eye? a least-square fit? ...)?
- please put the links to the ESA webpages in a footnote and make sure the formatting in latex does not break
- concerning the GAIA data: what is the difference between your approach and the one presented in Andrew+2022, MNRAS 516, 3? Their Figure 17 again shows a large number of low-q systems (when considering all samples apart from the 'gold' sample). Please clarify this.
- line 80:  please clarify if those binaries (also the ones mentioned in lines 85-87 and 89 and following) were cross-matched with the other catalogues mentioned here and whether potential duplicates were removed. Please also consider adding a table containing all the numbers and a brief explanation of the meaning of the different binary categories
- please clarify how many stars are in total in the mentioned interferometric binary catalogues, how does this relate to the number of dm-measurements? Also, why where these three catalogues chosen. Recent interferometric work on smaller samples, for example Klement+2022, ApJ 926, 2 or Lanthermann+2023, A&A 672, A6 detect a large number of low-q systems.

I did not mark anything related to the use of the English language but I encourage the author to revise this as well. Please also stick to common terms and notations (for example in line 143, 'matter transfer in the system' is usually referred to as 'binary mass transfer')

Author Response

Dear reviewer,

I thank you for extremely useful and constructive comments,
they greatly helped me to improve the paper.
I have made all required corrections and responded
all questions raised. All revisions
are highlighted (boldfaced) in the text. The details
of the revisions are given below, point-by-point.

Please, note that corrections required by other reviewers
are also made and highlighted in the text.

I am ready to answer your questions if you still have them.
Thank you very much again for your cooperation,
With best regards,
Oleg Malkov

General:
-----------------------
- in my opinion, the title of the paper is misleading as most of the paper is not about Gaia BH1. The paper provides a compilation of literature studies, summarising the distribution of mass ratios determined by different authors and methods. The paper demonstrates that few low-q binaries are known, however, that does not help to further our understanding, neither of low-q binaries nor of Gaia BH1

==> You are absolutely right, most of the paper is devoted to the q-distribution of binaries.
However, it was Gaia BH1 that triggered this study, and it is this object that I consider
very important for solving the discussed problem.

- while it is interesting and insightful to provide a compilation of measured mass ratios, please note that this is very much what is expected: low-q systems of two main-sequence stars are by definition harder to detect as their masses, and therefore their light contribution is very different. This is therefore nothing new, as you state in the beginning of Sect. 2, and thus the result of the paper is not surprising.

==> You are right: the lack of low-q systems among catalogued objects is nothing new.
In this work, I just demonstrate how (quantitatively) our ignorance extends,
and point out the importance of discovering and studying systems like Gaia BH (see the beginning of Sect. 3).

- several previous studies modelled the detection biases of their approaches, in particular towards low-q systems. For example, this was done extensively for spectroscopic studies (for example Sana+2013, A&A 550, A107; Bodensteiner+2021, A&A 652, A70; Banyard+2022, A&A 658, A69; Shenar+2022, A&A 665, A148; all demonstrating that the detection bias against low-q systems is high)

==> Thank you for the info, it is very valuable. I have added this information to the text (Sect. 2.1).

- while listing several methods detecting binaries, it remains unclear which methods were included and which ones were not. For example, high-contrast imaging has proven to be able to detect low-q systems (for example Pauwels+2023, arxiv:2307.13500 and references therin). Please clarify why this technique was not included, or consider including it.

==> I tried to consider the most "representative" observational types, that is,
those for which at least hundreds of binary systems are known (the exception is DLEB, Sect. 2.3,
I included this type due to the high accuracy of the parameters).
However, I read the Pauwels+2023 paper with great interest (thanks again for the helpful link)
and added the relevant fragment into the text (see page 5).

- along similar lines: for each of the methods, it remains unclear which catalogues / sources were considered and which ones were not. Please clarify this.

==> For each of the method I tried to use the most representative (largest) or
the most resent catalogue/source: SB9 for spectroscopic, ORB6 for orbiral, Torres+ for DLEB,
CEV for eclipsing, etc. The corresponding phrase has been added to the text (see page 5).

- concerning the figures: please sort them in a way that the first figure mentioned in the text is figure 1, and so on. Also, please include a legend (with the appropriate references when required) in each of the figures.

==> Done.

Specific:
-----------------------
Title: as mentioned above, I think the title should be changed to reflect the content of the paper better. Also, in my opinion, the 'q' should be written out in the title

==> I would still ask you to allow me to keep the title.

Abstract: please introduce 'q' as mass ratio

==> Done.

Introduction:
- please define q already here, and not only in Sect. 2

==> It is done (Introduction, line 18).

- please provide the error bars for m2 quoted here

==> Done.

- please always put units on numbers (and please stick to the common convenvtions, for example using M_sun instead of m_sun

==> Done.

Section 2:
- line 23: for BHs, this sentence does not hold: while the mass ratio is high, the BH does not contribute to the light at all. Given that the introduction discussed BH1, please clarify this statement such that it makes this clear.

==> Corrected.

- in my opinion, it would help the reader in their understanding to expand the second paragraph of Sect. 2 to explain what will be done in the subsequent subsections of the paper. Here, it would also be good to provide more detail what 'the main observational classes' are, and how they are defined (see my comment above on high-contrast imaging)

==> Done.

- please provide more information about why which set of catalogue / data was used for each of the subsections. For example, why was the SB9 catalogue considered for the spectroscopic binaries?

==> Done.

- line 56: you mention 'depth of the primary and secondary minimum' - do you mean the primary and secondary eclipse? (the same holds for line 68)

==> I do. Corrected.

- line 60: please clarify what exactly 'linear approximation' means, were the data including the error bars fitted? if yes, by which means (by eye? a least-square fit? ...)?

==> It is a least-squares best fit. Corrected.

- please put the links to the ESA webpages in a footnote and make sure the formatting in latex does not break

==> Done.

- concerning the GAIA data: what is the difference between your approach and the one presented in Andrew+2022, MNRAS 516, 3? Their Figure 17 again shows a large number of low-q systems (when considering all samples apart from the 'gold' sample). Please clarify this.

==> Thank you again for the useful reference. A brief discussion is added at the end of Sect. 2.5.

- line 80:  please clarify if those binaries (also the ones mentioned in lines 85-87 and 89 and following) were cross-matched with the other catalogues mentioned here and whether potential duplicates were removed. Please also consider adding a table containing all the numbers and a brief explanation of the meaning of the different binary categories

==> Those binaries were not cross-matched with the other catalogues. A brief explanation of the meaning of the different binary categories is added to the text.

- please clarify how many stars are in total in the mentioned interferometric binary catalogues, how does this relate to the number of dm-measurements? Also, why where these three catalogues chosen. Recent interferometric work on smaller samples, for example Klement+2022, ApJ 926, 2 or Lanthermann+2023, A&A 672, A6 detect a large number of low-q systems.

==> Usually, in interferometric catalogues, the number of measurements exceeds the number of studied objects
by 10-20% (for example, the Balega+ 2004 catalogue contains 111 measurements for 99 objects).
We did not pay close attention to this circumstance, since we cannot correctly build a q-distribution,
but we can only estimate the minimum q for a given catalogue.
For the same reason, we do not claim completeness of information about interferometric measurements.
However, thanks again for the valuable links. Klement+2022 investigate systems
that have undergone binary mass transfer (we try to exclude such systems from consideration),
and I have inserted a discussion of the work of Lanthermann+2023 in the text (end of Sect. 2.6).

Comments on the Quality of English Language

I did not mark anything related to the use of the English language but I encourage the author to revise this as well. Please also stick to common terms and notations (for example in line 143, 'matter transfer in the system' is usually referred to as 'binary mass transfer')

==> I checked my English again and made some corrections.

Reviewer 3 Report

This paper uses a variety of binary catalogues to estimate the q ratio distribution of discovered binaries and show that there are very few low-q systems known. This is linked to the recently discovered Gaia BH1 system, whose progenitor would be very difficult to discover with techniques available today. 

I find that the idea of the paper is interesting but there are some points that should be addressed before it can be considered for publication, especially considering the purposed link of missing low-q systems with Gaia BH1: 

- Black-hole binaries are not discussed in the paper although in low mass X-ray binaries mass ratios of 0.1 or below are quite typical. In this sense Gaia BH1 is not that special (the obvious difference is that the black hole in Gaia BH1 is not accreting due to its longer period). So I fail to see why Gaia BH1 is central to the argument of "missing low-q systems" as it seems to me the previous population of low-mass X-ray binaries could already be used for this claim. This point should be clarified in the paper. 

- Why is Gaia BH2 not mentioned in the paper?

- Is there a huge "low-q" population really missing due to observational bias or is the true mass ratio distribution decreasing sharply towards low q? Is there an answer to this question or clues based on the currently known binaries within the different catalogues? I don't find the existence of Gaia BH1 (which is one among hundreds of thousands of stars that were observed by Gaia within 500 pc) a convincing argument to claim that there is a huge number of missing low q systems. 

- There are more recent compilations of speckle interferometric binaries that have a larger number of low-q systems including several below 0.1 (e.g. Figure 9 in 2023AJ....165..180T) that are not mentioned and that could help in addressing the points above. 

- The author doesn't mention that Gaia-Hipparcos proper motion changes (e.g. 2021ApJS..254...42B) for nearby targets can detect small accelerations caused by objects as low mass as Jupiter-like planets. 

- line 19-20: in our understanding --of-- low-q systems 

- line 136: "cannot be overestimated": probably the author meant "underestimated"

Author Response

Dear reviewer,

I thank you for extremely useful and constructive comments,
they greatly helped me to improve the paper.
I have made all required corrections and responded
all questions raised. All revisions
are highlighted (boldfaced) in the text. The details
of the revisions are given below, point-by-point.

Please, note that corrections required by other reviewers
are also made and highlighted in the text.

I am ready to answer your questions if you still have them.
Thank you very much again for your cooperation,
With best regards,
Oleg Malkov

Comments and Suggestions for Authors

- Black-hole binaries are not discussed in the paper although in low mass X-ray binaries mass ratios of 0.1 or below are quite typical. In this sense Gaia BH1 is not that special (the obvious difference is that the black hole in Gaia BH1 is not accreting due to its longer period). So I fail to see why Gaia BH1 is central to the argument of "missing low-q systems" as it seems to me the previous population of low-mass X-ray binaries could already be used for this claim. This point should be clarified in the paper. 

==> Thank you very much for pointing out this my omission. You are right, there are dozens LMXBs with low q.
However, in our study we consider binary systems demonstrating _initial_ q-distribution,
i.e., without mass transfer. I have added a couple of phrases in the last but one paragraph of Sect.3.

- Why is Gaia BH2 not mentioned in the paper?

==> Thank you for the useful reference. I have added it in the text (see дшту 174).

- Is there a huge "low-q" population really missing due to observational bias or is the true mass ratio distribution decreasing sharply towards low q? Is there an answer to this question or clues based on the currently known binaries within the different catalogues? I don't find the existence of Gaia BH1 (which is one among hundreds of thousands of stars that were observed by Gaia within 500 pc) a convincing argument to claim that there is a huge number of missing low q systems. 

==> I fully agree with you. I do not think that Gaia BH1 is a convincing argument
to claim that there is a huge number of low q systems.
However, before Gaia BH, we did not know of _any_ system with such a small _initial_ value of q.

- There are more recent compilations of speckle interferometric binaries that have a larger number of low-q systems including several below 0.1 (e.g. Figure 9 in 2023AJ....165..180T) that are not mentioned and that could help in addressing the points above. 

==> Thank you once more for the useful reference. I have added a brief discussion in the text.

- The author doesn't mention that Gaia-Hipparcos proper motion changes (e.g. 2021ApJS..254...42B) for nearby targets can detect small accelerations caused by objects as low mass as Jupiter-like planets. 

==> Added to the text.

Comments on the Quality of English Language

- line 19-20: in our understanding --of-- low-q systems 

==> Corrected

- line 136: "cannot be overestimated": probably the author meant "underestimated"

==> Corrected

Reviewer 4 Report

Due to technical limitations, Low mass-ratio binaries is a kind of binary that are not fully studied and investigated in current binary catalog. The author investigate the mass ratio distributions of different kinds of binaries, studied the mass-temperature relations of double lined eclipsing binaries, and mass ratio-magnitude difference relations of main sequence binaries, present lists of interferometric binaries and a sample of very low mass-ratio binary. All these contents are useful information for low-mass ratio binaries. Although the low-mass ratio binaries is rare currently, the existence of these cases did indicates the incompleteness of our study on such kind of system, thus deserve more attention and studies.

Better to move Figure 3 to Figure 1.

Author Response

Dear reviewer,

I thank you for your comments.

I changed the order of the figures as you requested.
Please, note that corrections required by other reviewers
are also made and highlighted in the text.

I am ready to answer your questions if you still have them.
Thank you very much again for your cooperation,
With best regards,
Oleg Malkov

Round 2

Reviewer 2 Report

Dear author,

thank you very much for the updated version taking into account my comments. I have a few minor additional ones (I am not commenting on any corrections of the English).

1. I am okay with leaving the title as it is.

2. I would strongly recommend that you add a strong statement like the one you mentioned in your response ("In this work, I just demonstrate how (quantitatively) our ignorance extends,
and point out the importance of discovering and studying systems like Gaia BH (see the beginning of Sect. 3).") to the abstract of the manuscript itself.

3. line 32: I would rephrase: mainly focus on the observing techniques, which yield a large number...

4. please edit the text in lines 94-104. The formatting is off, and maybe it would be more easily readable if the information was given in a table.

5. Please consider adding to line 139: "high accuracy of the derived parameters"

I would recommend extensive language editing to improve the level of English and to make the manuscript more easily readable.

Author Response

Dear reviewer,

I thank you once more for extremely useful and constructive comments,
they greatly helped me to improve the paper.

>1. I am okay with leaving the title as it is.
>
>2. I would strongly recommend that you add a strong statement like the one you mentioned in your response ("In this work, I just demonstrate how (quantitatively) our ignorance extends,
>and point out the importance of discovering and studying systems like Gaia BH (see the beginning of Sect. 3).") to the abstract of the manuscript itself.
>
>3. line 32: I would rephrase: mainly focus on the observing techniques, which yield a large number...
>
>4. please edit the text in lines 94-104. The formatting is off, and maybe it would be more easily readable if the information was given in a table.
>
>5. Please consider adding to line 139: "high accuracy of the derived parameters"

Your recommendations 2-5 were adopted, and the text has been amended accordingly. 

>Comments on the Quality of English Language
>I would recommend extensive language editing to improve the level of English and to make the manuscript more easily readable.

I showed the manuscript to two native speakers, independently,
and they made a number of changes to the text.

I am ready to answer your questions if you still have them.
Thank you very much again for your cooperation,
With best regards,
Oleg Malkov

Reviewer 3 Report

The author has adequately addressed my comments and suggestions. 

The language could be improve for better legibility but it is understandable. 

Author Response

Dear reviewer,

I thank you once more for extremely useful and constructive comments,
they greatly helped me to improve the paper.

I showed the manuscript to two native speakers, independently,
and they made a number of changes to the text.

Thank you very much again for your cooperation,
With best regards,
Oleg Malkov